# Evaluation of the Governance Efficiency of Water Environmental Governance Efficiency in Yangtze River Delta from the Perspective of Multivariate Synergies

**DOI:** 10.3390/ijerph19042347

**Published:** 2022-02-18

**Authors:** Xiaoqiong Liu, Xu Wang, Feiyu Lu, Shuai Liu, Kunlun Chen

**Affiliations:** 1School of Environmental Studies, China University of Geosciences, Wuhan 430074, China; xiaoqiongliu@cug.edu.cn; 2China Institute of Mountaineering and Outdoor Sports, Wuhan 430074, China; 3School of Geography and Information Engineering, China University of Geosciences, Wuhan 430074, China; wangxu@cug.edu.cn; 4The Development Research Center of Kunshan Municipal People’s Government, Kunshan 215300, China; lfy0816@live.com; 5School of Arts and Communication, China University of Geosciences, Wuhan 430074, China; liushuai2020@cug.edu.cn; 6School of Physical Education, China University of Geosciences, Wuhan 430074, China

**Keywords:** water environment, pluralistic actors, governance efficiency, DPSIR–TOPSIS method, SNA method, Yangtze River Delta

## Abstract

Comprehensive governance of the watershed environment is one of the keys to urban and regional development and construction, which will affect not only the overall quality of urban economic development, but also the production and lives of urban residents. Since the economy in the Yangtze River Delta develops rapidly and the water environmental issues is more and more striking, it is in urgent need of moving forward the governance of water environment. This study empirically analyzes the governance efficiency of water environment in the Yangtze River Delta from 2006 to 2017 adopting the methods of the DPSIR (Driving Force-Pressure-State-Impact-Response Analysis model)–TOPSIS (Technique for Order Preference by Similarity to an Ideal Solution) and the SNA (Social Network Analysis) to clarify the roles and responsibilities of different cities and main contributors in the governance of water environment. According to the research, the following results are attained: first, due to the effects of pressure and the state subsystem, the Yangtze River Delta’s governance efficiency of water environment has increased steadily over time, from 0.3704 in 2006 to 0.4645 in 2017, but the disparities across cities have further widened. Second, in terms of contributors, the enterprises and governments play the main roles in the governance of water environment in recent years, while the public cannot always exert significant influence owing to unexpected environmental occurrences. Lastly, from the perspective of regional coordinated governance, the Yangtze River Delta resembles a tightly connected network of collaborative governance of water environment, with network connectivity and density growing year after year. However, the network structure of the governance efficiency of water environment in the study area is asymmetric, and network connectivity is higher inside the administrative regions, whereas spatial connectivity across provincial administrative boundaries has to be improved. The research scale and connotation in the field of the governance of water environment can be expanded and deepened through the study on the evaluation of the governance efficiency of water environment in the Yangtze River Delta, and it has considerable practical implications in modernizing the national governance system and capability.

## 1. Introduction

Since the reform and opening up of the 1970s, Chinese economy has greatly boomed with the rapid industrialization and urbanization advancing. However, the environment has faced increasing problems because of high-intensity and extensive development mode [1]. Among these problems, the most notable is water pollution, which causes annual social and economic losses of up to RMB 240 billion [2], making it a serious practical concern for China’s regional sustainable development [3]. Although China has promoted a series of relevant policy initiatives and institutional reforms to push forward the governance of water environment, it is far from sufficient [4,5]. Currently, the most pressing issue in the governance of water environment is determining how to increase governance efficiency and methods.

The governance of water environment and its efficiency have always been a hot researching point of many subjects. In the early studies, the focus is on the governance of point-source water environments, which is easy to identify and manage. Lots of environmental scientists measure the differences of treatment performance among the wastewater treatment plants by means of analyzing the changes in resource and energy inputs (lime volume, electricity consumption, etc.,), labor inputs, management and maintenance inputs, and widely recorded pollutant indexes (N, P, COD, etc.,) [6,7,8]. However, these studies put emphasis on the economic performance produced by technological treatment by the wastewater treatment plants in the water treatment, but they ignore the overall impact exerted by external factors.

The governance of nonpoint-source water environments at the urban and regional scales has evolved over times as new disciplines, such as geography and management, having focus on the governance of water environment. The governance efficiency of industry-specific water environments [9,10,11], as well as the integrated water environment at the urban and rural [12,13], administrative [14,15], and basin levels [13,16], were all investigated. The problem of nonpoint-source water pollution treatment at the urban and regional levels is more challenging due to the complexity of nonpoint-source water pollution [17]. At this scale, the governance of water environment entails not only technical aspects but also coordination among the government, enterprises, and the general public on how to achieve the optional governance form, resulting in a wider range of governance subjects and governance models.

The routes of various actors’ participation in the governance of water environment within administrative regions [18], as well as the role of multi-actor governance in enhancing the governance efficiency of water environment [19], have been studied by several academics. In terms of evaluation methodologies, most researchers used the DEA (data envelopment analysis) model to assess the governance efficiency of water environment [20,21,22], while others used the basic pollution index method [16] and the fuzzy comprehensive evaluation method [15]. The governance of water environment, on the other hand, is frequently linked to the population, economy, society, resources, and environment [23,24]. Existing studies applying the DEA model tend to have too many single indexes, and the relevant indicators are mostly engineering technical indicators involving the water environment itself (e.g., nitrogen and phosphorus in wastewater), with little attention paid to the structural factors influencing the governance efficiency of water environment, such as residents, enterprises, and other subjects. Furthermore, the inherent negative externalities of water environmental problems necessitate the construction of a cooperative governance mechanism [2], but there is no quantitative assessment of the water environment’s coordinated governance efficiency by diverse actors and regions. As far as the research scale is concerned, most previous studies focus on the national or provincial scale, and more in-depth studies on transboundary collaborative governance of the water environment at the economic zone level are needed.

The Yangtze River Delta is one of China’s most economically vibrant, innovative, and open regions, and it plays a critical role in the country’s modernization, participating in international governance, and the development of an ecological civilization system [25]. As a result of rapid urbanization and industrialization, the Yangtze River Delta’s water environment has been subjected to enormous pressure, and many water environmental indicators, such as the percentage of surface water cross-sections with water quality better than Class III, are far below the national standards [26]. Furthermore, because the Yangtze River Delta includes multiple provincial administrative districts, progress in cooperative governance of the regional water environment has been gradual, and water-related issues are numerous. Since the 18th National Congress, the government has paid increasing attention to the Yangtze River Delta’s biological environment, reflecting a shift in development concepts. The Yangtze River Delta Outline of the Integrated Regional Development Planning mandated in 2019 that the construction of a platform for monitoring the regional ecological environment and pollution sources be expedited, followed by the establishment of a joint mechanism for the coordinated governance of cross-regional and cross-basin water environments.

In this paper, the Yangtze River Delta is taken as a case. The following methods are adopted. The DPSIR analysis model is to construct a governance efficiency indicator system of water environment, the TOPSIS method and the SNA method are to conduct a comprehensive analysis of the governance efficiency of water environment in the study area from 2006 to 2017. Through this study, the changes in the study area are illustrated during the decade. The roles and responsibilities of different actors and cities in the governance efficiency of water environment are further clarified. In this context, the governance of water environment in the Yangtze River Delta can be improved collaboratively, which can contribute greatly to the modernization of the national governance system and governance ability.

## 2. Materials and Methods

### 2.1. Study Area

With a total water area of 24,500 km^2^, the Yangtze River Delta is China’s region with the highest density of river networks. It has more than 200 large and small lakes [27], as well as the Yangtze River, which is 425 km in length, and the canal, which is 819 km in length, and blessed with a pretty decent water environment. As the most economically developed region in China, the level of urbanization in the Yangtze River Delta has rapidly increased from 59.71% in 2006 to 70.78% in 2017. There are a huge number of industrial firms, especially chemical companies, that are also centered inside the Yangtze River Delta, and their wastewater discharge accounts for about 80% or more of the country’s total industrial waste discharge [26]. The sample cities chosen in this research are 15 core cities in the Yangtze River Delta, including Shanghai, Nanjing, Wuxi, Changzhou, Suzhou, Nantong, Yangzhou, Zhenjiang, Taizhou, Hangzhou, Ningbo, Jiaxing, Huzhou, Shaoxing, and Taizhou, due to data accessibility and comparability (Figure 1).

### 2.2. Research Methodology and Data Processing

#### 2.2.1. Driving Force-Pressure-State-Impact-Response Analysis Model (DPSIR)

The DPSIR analysis model is a type of conceptual model for index system evaluation that may look at how human and environmental socio-economic activities interact, including driving forces, pressure, state, impact, and responses [28,29].

The term “driving force” refers to the underlying influencing factors that may cause water environment changes, which include the two fundamental characteristics of nature and society [30], such as economic development status, population, natural resources, and environmental conditions. As a result, the five indicators X1–X5 (Table 1) have been chosen to characterize the economic, social, and natural driving factors in the governance of water environment. Driving forces can result in a variety of circumstances that put pressure on natural resources and the environment.

“Pressures” are the result of drivers’ actions, which are variables that put pressure on resources and the environment as a result of their actions, and have a more direct impact on the state of natural resources and the environment than drivers do. As a result, the direct pollutant discharge indicators in production and life, X6–X9 (Table 1), have been chosen as the pressure factors for the governance of water environment.

“State” refers to the state displayed by the water environmental system under pressure, which is a direct reflection of the evaluation goal of the governance of water environment. It is a direct representation of the evaluation objectives for the governance of water environment, and it is measured by X10 and X11 (Table 1).

“Impact” refers to the comprehensive impact on the ecological environment, the social economy, and other aspects caused by the changes in the above-mentioned factors. Studies have indicated that the governance efficiency of water environment has an important impact on the ecosystem, and traditional wastewater management’s high energy consumption may have a negative impact on the overall environmental benefits [31]. As a result, X12 (Table 1) represents the environmental impact of the governance of water environment, whereas X13, X14, and X15 (Table 1) represent the influence on economic development and people’s lives, respectively.

“Response” refers to the response measures taken to optimize the current state. Indexes like X16–X20 (Table 1) are chosen to reflect the response measurements of many subjects like the government, enterprises, and the public. The indexes are spilt by governance subjects in terms of responsibility sharing and beneficence, building on the work by Chen Shiyi et al. [32], to quantify the governance efficiency by different players.

#### 2.2.2. Technique for Order Preference by Similarity to an Ideal Solution method (TOPSIS)

TOPSIS (Technique for Order Preference by Similarity to an Ideal Solution) is a multi-criteria decision-making technique for ranking evaluation objects in relation to an ideal goal [33,34]. The main steps in the calculation are as follows:

(1) The raw data $x_{ij}\;(x_{ij}$ is the *jth* indicator of the *ith* city), they are standardized using extreme difference standardization to eliminate the effect of differing magnitudes, which is indicated as $x_{ij}^{\prime }$. To eliminate the effect of logarithm on the calculation, the standardized data are shifted to the right by 0.0001 units and recorded as $x_{ij}^{\prime \prime }$.
(1)$$x_{ij}^{\prime }=\frac{x_{ij}-\min\left(x_{1j},x_{2j},\ldots ,x_{nj}\right)}{\max\left(x_{1j},x_{2j},\ldots ,x_{nj}\right)-\min\left(x_{1j},x_{2j},\ldots ,x_{nj}\right)}\;(\mathrm{if}\;x_{ij}\;\mathrm{is}\;a\;\mathrm{positive}\;\mathrm{indicator})$$
(2)$$x_{ij}^{\prime }=\frac{\max\left(x_{1j},x_{2j},\ldots ,x_{nj}\right)-x_{ij}}{\max\left(x_{1j},x_{2j},\ldots ,x_{nj}\right)-\min\left(x_{1j},x_{2j},\ldots ,x_{nj}\right)}\;(\mathrm{if}\;x_{ij}\;\mathrm{is}\;a\;\mathrm{negative}\;\mathrm{indicator})$$
(3)$$x_{ij}^{\prime \prime }=x_{ij}^{\prime }+0.00001$$

(2) The index weights are calculated by combining the entropy weight method (EWM) with the mean squared deviation (MSD) (Table 2).

a. The entropy weight method (EWM) is an objective assignment method that calculates the entropy weight of each indicator based on the degree of variation of each indicator using information entropy, and then corrects the weight of each indicator by obtaining the weight of the indicator.

First, calculate the weight of the *ith* item’s indicator value under the *jth* indicator:(4)$$d_{ij}=x_{ij}^{\prime \prime }/\overset{n}{\underset{i=1}{\sum }}x_{ij}^{\prime \prime }$$

Next, calculate the *jth* indicator’s entropy value:(5)$$e_{j}=-\frac{1}{lnn}\cdot \overset{n}{\underset{i=1}{\sum }}d_{ij}lnd_{ij}$$

Finally, the weights of each indicator are calculated by information entropy:(6)$$w_{j}=\left(1-e_{j}\right)/\overset{m}{\underset{j=1}{\sum }}\left(1-e_{j}\right)$$

b. The mean squared deviation (MSD) method is an objective assignment method for determining the weight coefficient, which reflects the random variable’s degree of dispersion, and its most essential and commonly used indicator is the amount of the mean squared deviation of this random variable. Each of the three criterion layers’ evaluation indicators is treated as a random variable in this study, and the mean squared deviation of each indicator random variable is calculated, then the single mean squared deviation is normalized, yielding the weight coefficient of each evaluation indicator. The following are the steps in the calculation:

First, calculate the mean value of the variables:(7)$$E(Q_{i})=-\frac{1}{n}\cdot \overset{n}{\underset{i=1}{\sum }}x_{ij}^{\prime \prime }$$

Next, the mean squared deviation is calculated:(8)$$\sigma (Q_{i})=\sqrt{\overset{n}{\underset{i=1}{\sum }}(x_{ij}^{\prime \prime }-E(Q_{i}){)}^{2}}$$

Finally, the weight coefficients are calculated:(9)$$w_{j}=\sigma (Q_{i})/\overset{n}{\underset{i=1}{\sum }}\sigma (Q_{i})$$

(3) The weighted decision matrix (R) is created by using the aggregate weight (AW).
(10)AW=(EWM+MSD)/2
(11)$$R=\left(r_{ij}\right)\;,\;r_{ij}=aw_{j}*x_{ij}^{\prime }$$
where, $aw_{j}$ is the AW of the *jth* indicator, $x_{ij}^{\prime }$ is the normalized *jth* index of the city *i*.

(4) Determination of the ideal solutions. The ideal solutions are divided into positive ideal solution ($S_{j}^{+}$) and negative ideal solution ($S_{j}^{-}$).
(12)$$S_{j}^{+}=\max\left(r_{1j},r_{2j},\cdot \cdot \cdot ,r_{nj}\right)$$
(13)$$S_{j}^{-}=\min\left(r_{1j},r_{2j},\cdot \cdot \cdot ,r_{nj}\right)$$

(5) Distances between various indicators of the governance efficiency of water environment and positive and negative ideal solutions are calculated.
(14)$$D_{i}^{+}=\sqrt{\sum_{j=1}^{n}{\left(S_{j}^{+}-r_{ij}\right)}^{2}}$$
(15)$$D_{i}^{-}=\sqrt{\sum_{j=1}^{n}{\left(S_{j}^{-}-r_{ij}\right)}^{2}}$$

(6) Calculation of the relative degree of closeness $C_{i}$ of each index, that is, the governance efficiency of water environment.
(16)$$C_{i}=\frac{D_{i}^{-}}{D_{i}^{+}+D_{i}^{-}}$$

#### 2.2.3. Social Network Analysis (SNA)

Social network analysis (SNA) is a research method that explores the attributes of association relationships between network nodes and their structural properties, and the construction of a network incidence matrix is the foundation of social network analysis. To analyze the governance of water environment within an urban cluster using the SNA method, the first step is to create a governance association network of water environment between cities in the cluster. The gravity model is a standard way of determining the association between distinct regions, and it works like this:(17)$$F_{ij}=\frac{m_{i}m_{j}}{d_{ij}^{2}}K$$

$F_{ij}$ denotes the gravity values of the governance of water environment of cities *i*, *j*; $m_{i}$ and $m_{j}$ denote the efficiency values of the governance of water environment of cities *i*, *j*; $d_{ij}$ denotes the geographical distance between cities *i* and *j*; and *K* denotes the gravitational constant, which is generally taken as 1. θ is the incidence network’s threshold, and *E* is the gravity matrix’s mean value.

The gravitational values of the 15 prefecture-level cities in the Yangtze River Delta city cluster are calculated two by two to form a 15 × 15 gravitational matrix. A sufficient threshold is often chosen to binarize the correlation matrix and translate the gravitational matrix into a network correlation matrix in order to enable network analysis and filter out the influence of relatively weak connections [35,36]. The network threshold θ is compared to the gravitational values in the gravitational matrix. If $F_{ij}$ is greater than θ, there is a correlation between the two cities’ governance of water environment, which is recorded as 1, and vice versa, which is recorded as 0.
(18)$$\theta =\left[E+\min\left(F_{ij}\right)\right]/2$$
where θ is the correlation network’s threshold value, and E is the gravity matrix’s mean value.

The SNA often exposes broad network properties like network density, network relatedness, network hierarchy, and network efficiency, as well as network characteristics like the degree centrality, betweenness centrality, closeness centrality, and system centrality for each node [37,38,39,40,41,42,43,44].

In Equation (19), *m* denotes the number of actual relationships in the relationship matrix, n is the number of nodes, and *n* × (*n* − 1) is the maximum number of relationships for any two nodes connected. The deeper the link between the nodes, the higher the value of this indicator, which shows the closeness of the network structure.
(19)$$\mathrm{network}\;\mathrm{density}=\frac{m}{n\left(n-1\right)}$$

In Equation (20), *r* is the number of unreachable pairs of points in the network, or the number of pairs of points that are not directly or indirectly associated between the two nodes. This indicator represents the network structure’s robustness and vulnerability. The network structure has a high degree of association and is more robust when any two nodes are coupled with each other.
(20)$$\mathrm{network}\;\mathrm{relatedness}=1-\frac{r}{n\left(n-1\right)/2}$$

In Equation (21), *s* is the number of symmetrically reachable relationship pairs in the relationship matrix, while *max*(*s*) is the maximum possible symmetrically reachable relationship pairs in the network. This indicator measures the degree of asymmetric reachability among network nodes; a higher value implies that the hierarchical structure among nodes is more unequal, and a few nodes hold a dominant position in the network structure.
(21)$$\mathrm{network}\;\mathrm{hierarchy}=1-\frac{s}{max\left(s\right)}$$

In Equation (22), *v* is the number of redundant lines in the network and *max*(*v*) is the maximum possible number of redundant lines in the network. This indicator represents the number of redundant lines in the network. The lower the network efficiency, the more external connections the nodes have, the more overlaps there are, and the more stable the network is.
(22)$$\mathrm{network}\;\mathrm{efficiency}=1-\frac{v}{max\left(v\right)}$$

In Equation (23), *d_i_* is the number of direct relationships that exist between node *i* and other nodes, and *n* is the number of nodes. The bigger the value of this indicator, the more connections the node has with other nodes, and the larger the value of the indicator, the more central the node is in the network.
(23)$$\mathrm{degree}\;\mathrm{centrality}=1-\frac{d_{i}}{n-1}$$

In Equation (24), *g_jk_* is the number of relational paths that exist between nodes *j* and *k*, and *g_jk_*(*i*) is the number of relational paths between nodes *j* and *k* to pass through node *i*. The greater the value of the indicator, the more important node *i*’s role in the network as an intermediary node, as it is at the node where many nodes are connected to each other.
(24)$$\mathrm{betweenness}\;\mathrm{centrality}=\frac{\sum_{j<k}g_{jk}\left(i\right)/g_{jk}}{\left(n-1\right)\left(n-2\right)}$$

In Equation (25), *d_ij_* represents the shortcut distance between nodes *i* and *j*. The value is taken as 1 if the two cities are directly connected and not indirectly connected through other cities. The higher the indicator’s value, the better the exchange path’s accessibility, suggesting the presence of direct connections between the node and many other nodes.
(25)$$\mathrm{clossness}\;\mathrm{centrality}=\overset{n}{\underset{j=1}{\sum }}d_{ij}$$

In Equation (26), the centrality indicators are considered together using system centrality.
(26)$$system\;centrality=\frac{\left(degree\;centrality+betweenness\;centrality+clossness\;centrality\right)}{3}$$

### 2.3. Data Sources

The evaluation system’s data originate from the China City Statistical Yearbook, the City Statistical Yearbooks of 15 prefecture—level cities, the Water Resources Bulletin, the Environmental Status Bulletin, the EPS database (Global Statistical Data Analysis Platform) (https://www.epsnet.com.cn/index.html#/Index) (accessed on 11 November 2021), and the Baidu Index, with some missing data imputed.

## 3. Results

### 3.1. DPSIR–TOPSIS-Based Comprehensive Evaluation of the Governance Efficiency of Water Environment

#### 3.1.1. Analysis of DPSIR Subsystem

The driving force index shows an overall upward trend from 2006 to 2017, with an average annual growth rate of 2.64%, indicating that the driving force of the driving forces subsystem on the Yangtze River Delta’s governance efficiency of water environment has been improving year by year. The total water resources in the 15 sample cities in the Yangtze River Delta all declined to various levels in 2017, resulting in a major reduction in the driving force. As a result of rapid socio-economic development and urbanization, the Yangtze River Delta has steadily transitioned from extensive growth based on high-energy-consuming sectors to green development based on high-tech industries and green industries [45]. Increased demands for the governance of water environment from all walks of life have resulted from the alteration of development concepts and models, boosting their efficiency.

During the research period, the overall pressure index remained high and stable, with an average annual increase rate of only 0.73%, which contributed the most to the governance efficiency of water environment. Because the diverse pressure index is negative for the improvement of the water environment, the decrease in the value of the pressure will help to lower the pressure and improve the governance efficiency of water environment in the Yangtze River Delta. Since the development of a two-oriented society in 2006, the government has enacted policies and regulations that place rigorous limits on the discharge and utilization of industrial waste [32,46], owing to the continued rise in the importance of water quality in all walks of life. As a result, the total industrial waste discharge of the 15 sample cities in the Yangtze River Delta has decreased drastically, from 4.537 billion tons in 2006 to 2.514 billion tons in 2017, significantly improving the Yangtze River Delta’s governance efficiency of water environment.

The status subsystem’s growth rate was the highest during the research period, notably since 2015, when it topped 8%, having a striking influence on the improvement of the governance efficiency of water environment. During the research period, the Yangtze River Delta Water Function Zone’s water quality compliance rate and the proportion of surface water sections with water quality better than Grade III both significantly improved, indicating that the Yangtze River Delta’s water quality has significantly improved, which directly reflects the governance of water environment.

The overall impact index fluctuated slightly around 0.38 over the 12-year period due to the impact subsystem. Despite the water environment in the Yangtze River Delta facing numerous challenges under driving forces and pressure, the impact of water pollution on the ecological environment, the economy, and society was relatively stable, owing to the efforts of the government, enterprises, and the public from all walks of life.

The average annual growth rate of the response index in the response subsystem was 4.06%, indicating the beneficial role played by diverse actors in improving the quality of the water environment and improving the governance efficiency of water environment (Figure 2). In comparison to 2006, the government’s efforts to improve the governance efficiency of water environment have increased the centralized governance rate of wastewater governance plants, the density of drainage pipes in built-up areas, and the green coverage rate of built-up areas by 15.30%, 4.51 km/km^2^, and 2.43%, respectively. When compared to 2006, the average rate of industrial water reuse has increased by 23.12%, demonstrating that industrial enterprises are continuing to invest in technology, enhance production techniques, and apply cleaner production methods. The Baidu index of water pollution in these Yangtze River Delta sample cities was twice as high as in 2006, indicating that public environmental awareness has improved greatly and engagement in the governance of water environment has increased visibly in the Yangtze River Delta.

#### 3.1.2. Overall Evaluation of the Governance Efficiency of Water Environment

The governance efficiency index of water environment for the cities in the Yangtze River Delta was generated for each calendar year, and Table 3 provides the typical indexes for the first and last years, as well as the midway year.

In summary, the governance efficiency of water environment in the Yangtze River Delta grew consistently over the study period, from 0.3704 in 2006 to 0.4645 in 2017, with an average annual growth rate of 2.13%, delivering an overall jump from a low to a higher level. The majority of the 15 sample cities had low and lower levels of the governance efficiency of water environment in 2006, and none had a high level. After more than ten years of expansion, the Yangtze River Delta’s governance efficiency of water environment has substantially improved. By 2017, more than half of the sample cities had achieved a medium or high level of the governance efficiency of water environment, with only Jiaxing still below the baseline grade.

In terms of the spatial dimension, the Yangtze River Delta cities’ governance efficiency of water environment has evolved in a heterogeneous manner, and the gap in governance efficiency across cities has widened. As of 2017, Shanghai and Hangzhou are the two cities with high levels of the governance efficiency of water environment, and they are leading the average progress in this area. Jiaxing, Taizhou, and Zhenjiang, in contrast, are under the dual influence of a poor water environmental foundation and slow governance efficiency improvement, and their governance efficiency of water environment is always at a low level, and the gap with the Yangtze River Delta’s average level of the governance efficiency of water environment is widening, leaving more room for improvement in the future. Since the inception of “12th Five-Year Plan” for national development, the Yangtze River Delta has been strengthening its efforts for the industrial restructuring and the elimination of backward production, high-energy-consumption and high-pollution enterprises have been steadily pushed to the periphery cities and the pressure on the water environment in the area where the secondary industry is concentrated has been increasing [47]. As a result, this area cannot improve its governance ability of water environment and the level of regional cooperative governance of water environment in the short term. Rapidly raising the level of the governance of water environment in areas where the secondary industry is concentrated and the level of the regional governance of water environment is difficult, resulting in various characteristics of the Yangtze River Delta’s development level of the governance efficiency of water environment.

### 3.2. Multi-Coordination Analysis of the Governance Efficiency of Water Environment

#### 3.2.1. Analysis of Multiple Actors for the Governance of Water Environment

Figure 3 depicts the findings of the calculation of the governance efficiency of water environment for various actors.

Although the government did not contribute as much to environmental protection before 2015, this condition improved greatly after that. In particular, compared to 2008, the governance efficiency of water environment declined by 14.03% in 2009. On the one hand, the Yangtze River Delta’s GDP growth rate has slowed as a result of the 2008 financial crisis; on the other hand, due to the obvious increase in waste discharge, the centralized governance rate of sewage governance plants in the Yangtze River Delta fell by 8.33% on average in 2009 (except Hangzhou, Jiaxing, Huzhou, and Taizhou), and the government’s governance efficiency has declined significantly under these integrated endeavors (Figure 3).

Enterprises’ governance efficiency of water environment has been progressively improving. In the early stages of the project, the enterprises were generally at a modest level among the three participants. After 2012, various firms in the Yangtze River Delta have promoted cleaner production and improved the water resource utilization rate to reduce pollution emissions; on the other hand, they have gradually advanced the transformation and upgrading toward high technology, thus pushing forward the continuous optimization of the Yangtze River Delta’s industrial structure. The governance efficiency of enterprises is improving continually, and it is gradually taking the lead in the Yangtze River Delta’s governance of water environment.

The Yangtze River Delta’s governance of water environment is heavily influenced by the public. As eco-social development and information accessibility improved, and substantial governance results were attained, the public played an active role in the governance of the water environment. In general, their governance efficiency of water environment is improving. Sudden environmental events, on the other hand, affect the public’s water environment because of the deep intuitive feeling of water environmental problems and the characteristics of a greater health threat in the short term [48], and there is a positive relationship between the frequency of sudden water pollution incidents in related studies [49].

#### 3.2.2. Analysis of Coordinated Governance of Water Environment in Yangtze River Delta


(1)Overall Structural Characteristics of the Network


The number of spatial associations and network density of the governance of water environment in the Yangtze River Delta has been gradually increasing year after year. During the research period, the two increased by 47.06% and 46.91%, respectively, demonstrating that the spatial connections of the governance of water environment among cities in the Yangtze River Delta are becoming increasingly essential (Figure 4). The total efficiency of the association network is depicted in Figure 5, and it shows a fluctuating downward trend. At the same time, by referring to the calculation results during the research period, when the network correlation degree is 1 and the network grade degree is 0, we can find that the trend of the spatial correlation network in the Yangtze River Delta is fluctuant and increasing, and the proportion of two-way spillover spatial connections between the governance of water environment in each prefecture-level city has risen.(2)Individual Centrality Analysis of the Network

According to the calculation results, we can find that the mean value of degree centrality of the governance efficiency of water environment in the study area gradually increased from 40.48 to 59.52 during the research period. For the resultant values of eight cities, Hangzhou, Shaoxing, Taizhou, and Shanghai were higher than the value in 2017, which indicates that these cities have more connections in the network and are close to the network’s relative center. While Nanjing, Changzhou, Yangzhou, Taizhou, and other cities ranked behind, having few connections with other cities, and are in the peripheral part of the network (Figure 6).

The trend of mean closeness centrality of the governance efficiency of water environment in the Yangtze River Delta is consistent with the trend of mean degree centrality, rising from 60.54 in 2006 to 72.01 in 2017, it means that the Yangtze River Delta’s direct linkage to water pollution is steadily increasing and the link is very close. Hangzhou, Shaoxing, Shanghai, and other four cities have a higher rating than the city cluster’s average, suggesting that these seven cities can create direct connections with other cities in the governance network of water environment more swiftly. Correspondingly, the centrality values of the eight cities as Taizhou, Nanjing, Yangzhou, and Changzhou are below the mean and are considered as slow movers at the network’s edge (Figure 6).

In contrast to the preceding two indexes, both the total and average values of the betweenness centrality of the governance efficiency of water environment in the Yangtze River Delta have dropped. The total value is from 144 to 86 while the average is from 9.60 to 5.73. What can explain the result is that two cities can build a direct win–win cooperation to promote the governance of water environment without the third party’s coordination and risk sharing. As a result, the transaction cost can be saved and the efficiency is improved. From the research, we also can find that Hangzhou, Shaoxing, Huzhou, Shanghai, and other cities can serve as intermediary centers in the Yangtze River Delta. The governance of water pollution in these cities have a strong and dominated impact on the governance in other cities. While the intermediate center degree is far below the national average, which shows that these cities have limited ability to manage the flow of resources and information (Figure 6).

From the view of a more integrated system centrality, the mean and standard deviation of system centrality of the governance efficiency of water environment in the sample cities from 2006 to 2017 are decreasing, presenting the trend of decentralization with the governance cooperation of water environment advancing and win–win cooperation promoting. Apart from it, for the governance linkage network of water environment, the Yangtze River Delta is structured with asymmetrical features. What can also be found that the coordinated governance linkage of water environment among the cities in the same provinces is strong [50] but weak among the cities in different provinces. This is in accordance with the study of Tian Yuanhong and so on [4]. In view of the Yangtze River Delta’s governance linkage network of water environment, the prefecture-level cities in northern Zhejiang province and Shanghai have a relatively higher system centrality, which is consistent with the findings of Zhou Fengqi and other scholars from the perspective of the provincial scale in the Yangtze River Delta [49]. Nowadays, the cities in the southern Jiangsu province have improved the governance efficiency of water environment greatly, however, the system centrality of these cities is generally low due to its centralized governance model driven by central funding (Figure 6).

## 4. Conclusions

This comprehensive analysis of the efficiency of governance water environment provided some reference value for explaining the roles and duties of various actors and regions in the governance of water environment, enhancing it in China, and perfecting the model. The following are the key conclusions of this study:

(1) The DPSIR analysis model is based on the integrated analysis of the dynamic mechanisms of the governance of water environment, allowing for a more thorough assessment of the governance efficiency of water environment. The driving forces subsystem, as potential causes of risks and changes, often showed an increasing trend during the research period, and the driving effects on enhancing the Yangtze River Delta’s governance efficiency of water environment improved year by year. As the more direct pressure indicator, the pressure subsystem contributes the most to the governance efficiency of water environment. The state subsystem is a visual representation of governance efficiency of water environment. Clearly, the index growth rate has a major impact on the governance efficiency of water environment. The impact subsystem index has been generally stable throughout time, indicating that the impact of water pollution on the ecological environment, economy, and society has remained relatively constant. The average annual growth rate of the response subsystem index is 4.06%, suggesting that the response measures from all sectors of society have played a substantial role in enhancing the governance efficiency of water environment. In general, the Yangtze River Delta’s governance efficiency of water environment has increased steadily over time, and the governance efficiency of water environment has risen from a low to a reasonable high level. It does, however, have a varied evolutionary spatial characteris-tic, and the gap in the governance efficiency of water environment has grown even wider.

(2) The governance efficiency of different actors in terms of water environment differs noticeably. The government’s governance efficiency of water environment was initially low in the early stages, but it has substantially improved since 2015. Enterprise’s governance efficiency has constantly improved, and it has increasingly assumed a leading role in the Yangtze River Delta’s governance of water environment. The public’s role in the Yangtze River Delta’s governance of water environment has been steadily increasing. However, sudden environmental occurrences with particular swings have an impact on their governance efficiency.

(3) At the regional level of coordinated governance, the Yangtze River Delta’s spatial association of the governance of water environment is growing year by year, displaying the characteristics of a multi-city, multi-threaded, cross-regionally closely linked collaborative governance network, but with asymmetric structural characteristics. The degree of network incidence within each administrative region is high, especially in Zhejiang and Shanghai, which are relatively more central in the research area’s collaborative governance network of water environment, and governance coordination of water environment across provincial administrative boundaries still needs to be strengthened.

## 5. Reflections

(1) To track the governance efficiency of water environment [24], it is important to develop a scientific and sufficient index system. However, according to the current research literature, local and international scholars choose indicators differently, and no general consensus has been reached. The governance of water environment is influenced by a complicated and diverse set of factors, including population, economics, society, and resources, all of which are interconnected. On the basis of summarizing relevant research and combining the actual situation of the Yangtze River Delta, this paper draws on the outstanding advantages of the DPSIR model in reflecting the interaction among social economy, environmental governance, and governance policies in the governance of water environment. Based on the principles of comparability, comparability, and accessibility, an evaluation index system with a total of 20 indicators is constructed at three levels, including the target layer, the dimension layer, and the indicator layer. Compared with similar studies in China, we pay more attention to the structural factors of the governance efficiency of water environment when selecting indicators, and comprehensively consider many indicators involved in contributors including the government, enterprises, and the public (Table 1).

(2) The governance of water environment is characterized by externality, in which the interests of the government, enterprises, the public, and other stakeholders collide. The governance of water environment in urban clusters is characterized by complexity, extensibility, and time lag, as well as a complex network structure with various subjects, multiple levels, and multiple disciplines. The SNA method is a quantitative analysis method that combines graph theory and mathematical models to study the association between social actors, and it has been widely utilized in recent years to examine the complex relationship structure between regions. It is relatively rare in China to apply the SNA method to issues which are related to the governance of water environment. From the research results, it can effectively depict the complicated network structure of the governance of water environment inside urban clusters.

(3) This paper constructs a governance efficiency evaluation system of water environment based on the DPSIR analysis model, and uses the TOPSIS evaluation method and SNA method to conduct a comprehensive analysis of the governance efficiency of water environment in the Yangtze River Delta from 2006 to 2017. Theoretically, studies on the governance of water environment at the scale of watersheds or economic zones can effectively broaden the scope of current water environmental research. Meanwhile, multidimensional and collaborative perspective can serve to extend the content of the traditional research frameworks of the governance of water environment and help improve environmental governance-related theories. In practice, assessing the governance efficiency of water environment in the Yangtze River Delta will be helpful for clarifying the roles and responsibilities of various actors and regions in the governance of water environment, addressing issues such as information asymmetry, promoting coordinated governance in the Yangtze River Delta, and advancing the modernization of the national governance system and ability.

(4) Based on the preceding conclusions and in conjunction with current conditions in the Yangtze River Delta, the following enlightenments are reached: ① It is important to keep the concept of systemic governance alive. The connection between socio-economic activity and the natural environment is reflected in the state of the water environment. The Yangtze River Delta’s governance of water environment should consider regional economic development, population, natural resources, and environmental circumstances. ② To play a leadership and supervisory role in water protection, the government must further improve the governance efficiency of water environment. Meanwhile, the governance efficiency index of water environment should be fully utilized to ensure the right to know and supervise the actors involved, such as enterprises and the public, and to develop the governance system of water environment through multi-participation. ③ At this stage, we need to pay more attention to the water environmental problems of small and medium-sized cities on the Yangtze River Delta’s outskirts, continue to improve the system and mechanisms of the coordinated governance of water environment in the research area, and strive to solve the problem of insufficient water environmental cooperation across provincial administrative regions.

## Figures and Tables

**Figure 1 ijerph-19-02347-f001:**
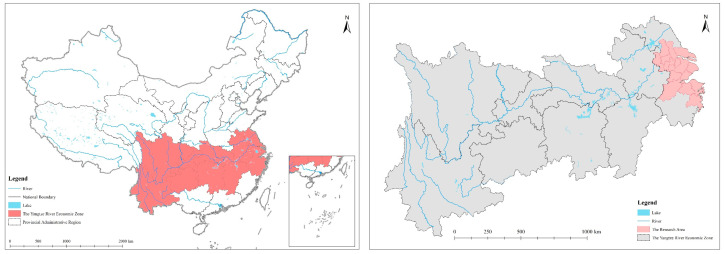
Location of the research area (The map on the left presents the relative position of the Yangtze River Economic Belt in China, while the map on the right presents the relative position of the Yangtze River Delta in the Yangtze River Economic Belt).

**Figure 2 ijerph-19-02347-f002:**
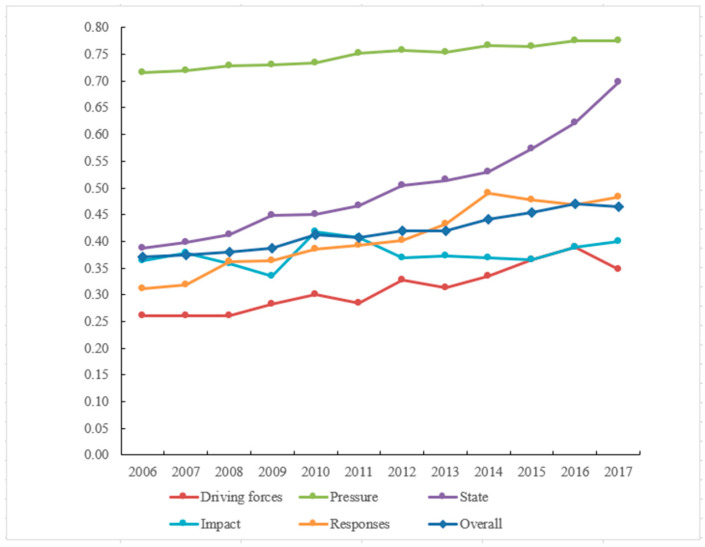
DPSIR subsystem evaluation indicators of Yangtze River Delta.

**Figure 3 ijerph-19-02347-f003:**
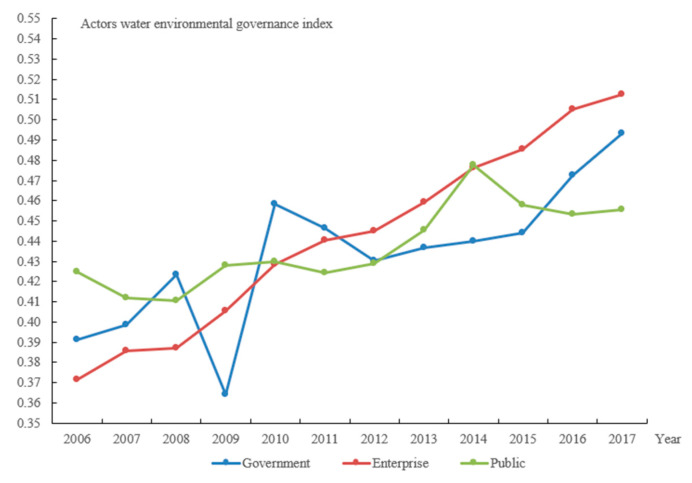
Governance index of water environment for synergistic actors in the Yangtze River Delta.

**Figure 4 ijerph-19-02347-f004:**
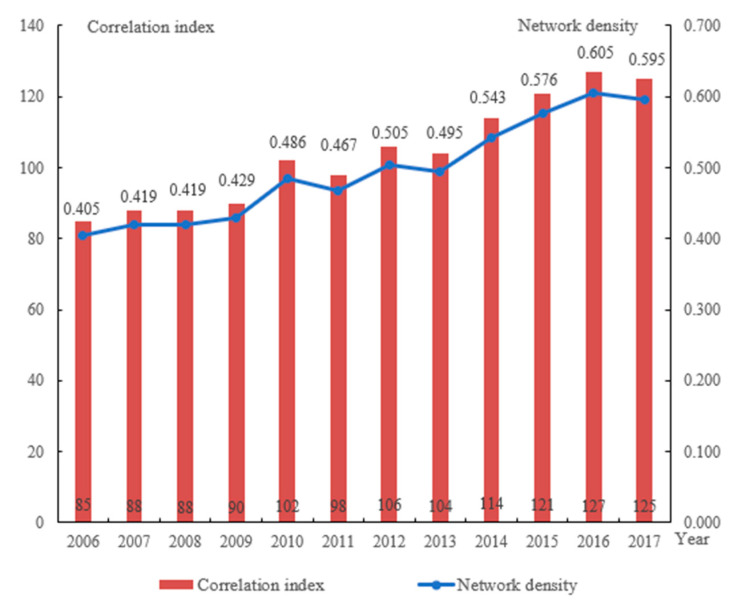
Correlation index and network density of the governance of water environment of Yangtze River Delta in 2006–2017.

**Figure 5 ijerph-19-02347-f005:**
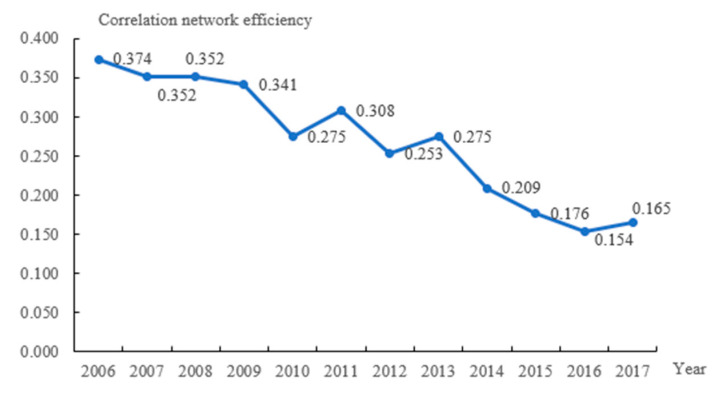
Correlation network efficiency of Yangtze River Delta’s governance of water environment in 2006–2017.

**Figure 6 ijerph-19-02347-f006:**
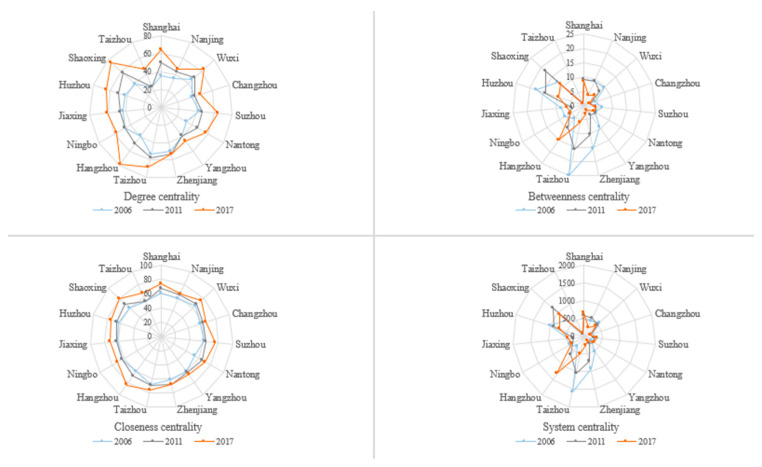
Individual centrality of the governance network of water environment in the Yangtze River Delta in 2006–2017.

**Table 1 ijerph-19-02347-t001:** Index system of the governance efficiency of water environmental in the Yangtze River Delta.

Target Layer	Dimension Layer	Index Layer	Index Direction	Actor
Index System of the Governance Efficiency of Water environmental in the Yangtze River Delta	Driving forces D	per capita GDP (X1)	+	Government
		Percentage of tertiary industry (X2)	+	Enterprise
		Urbanization level (X3)	+	Government
		Population density (X4)	+	Public
		Total amount of water resources (X5)	+	Common
	Pressure P	Industrial wastewater discharge (X6)	_	Enterprise
		Application of agricultural fertilizers (X7)	_	Enterprise
		Residential water consumption (X8)	_	Public
		Discharge of urban domestic sewage (X9)	_	Public
	State S	Conformity rate of water quality in water function areas (X10)	+	Common
		Percentage of cross section of surface water quality better than III (X11)	+	Common
	Impact I	Comprehensive energy consumption of water production and supply industry (X12)	_	Enterprise
		GPD growth rate (X13)	+	Government
		Total profit of industrial enterprises (X14)	+	Enterprise
		Residents’ healthcare spending (X15)	_	Public
	Responses R	Governance rate of sewage governance plant (X16)	+	Government
		Density of drainage pipes in built-up areas (X17)	+	Government
		Greening coverage rate of built-up areas (X18)	+	Government
		Industrial water reuse rate (X19)	+	Enterprise
		Baidu index of water pollution (X20)	+	Public

**Table 2 ijerph-19-02347-t002:** Calculation results of EWM method and MSD method.

Target Layer	Dimension Layer	Index Layer	EWM	MSD	AW
Index system of the Governance Efficiency of Water environment in the Yangtze River Delta	Driving forces D	per capita GDP (X1)	0.044	0.029	0.036
		Percentage of tertiary industry (X2)	0.067	0.052	0.060
		Urbanization level (X3)	0.057	0.059	0.058
		Population density (X4)	0.108	0.058	0.083
		Total amount of water resources (X5)	0.149	0.045	0.097
	Pressure P	Industrial wastewater discharge (X6)	0.026	0.058	0.042
		Application of agricultural fertilizers (X7)	0.027	0.056	0.042
		Residential water consumption (X8)	0.024	0.056	0.040
		Discharge of urban domestic sewage (X9)	0.017	0.050	0.034
	State S	Conformity rate of water quality in water function areas (X10)	0.064	0.062	0.063
		Percentage of cross section of surface water quality better than III (X11)	0.041	0.057	0.049
	Impact I	Comprehensive energy consumption of water production and supply industry (X12)	0.022	0.055	0.038
		GPD growth rate (X13)	0.024	0.056	0.040
		Total profit of industrial enterprises (X14)	0.134	0.043	0.089
		Residents’ healthcare spending (X15)	0.006	0.033	0.020
	Responses R	Governance rate of sewage governance plant (X16)	0.016	0.051	0.033
		Density of drainage pipes in built-up areas (X17)	0.053	0.047	0.050
		Greening coverage rate of built-up areas (X18)	0.008	0.030	0.019
		Industrial water reuse rate (X19)	0.014	0.050	0.032
		Baidu index of water pollution (X20)	0.100	0.053	0.077

**Table 3 ijerph-19-02347-t003:** Governance efficiency index and evolution for water environment of Yangtze River Delta.

	2006		2011		2017		Average Annual Growth Rate of the Governance Efficiency of Water Environment
	Evaluation Value	Grade	Evaluation Value	Grade	Evaluation Value	Grade	
Shanghai	0.3705	Lower	0.4866	Higher	0.5738	High	4.06%
Nanjing	0.3553	Low	0.4109	Medium	0.4709	Higher	2.59%
Wuxi	0.3498	Low	0.4358	Medium	0.4700	Higher	2.72%
Changzhou	0.4038	Medium	0.4127	Medium	0.4267	Medium	0.50%
Suzhou	0.4020	Medium	0.4002	Medium	0.5060	Higher	2.11%
Nantong	0.3295	Low	0.3851	Lower	0.4704	Higher	3.29%
Yangzhou	0.3649	Lower	0.3917	Lower	0.4203	Medium	1.29%
Zhenjiang	0.3697	Lower	0.3920	Lower	0.4222	Medium	1.22%
Taizhou	0.3510	Low	0.3757	Lower	0.4252	Medium	1.76%
Hangzhou	0.4036	Medium	0.4957	Higher	0.6004	High	3.68%
Ningbo	0.3849	Lower	0.3958	Medium	0.5070	Higher	2.54%
Jiaxing	0.3253	Low	0.3452	Low	0.3729	Lower	1.25%
Huzhou	0.3740	Lower	0.3984	Medium	0.4251	Medium	1.17%
Shaoxing	0.4028	Medium	0.4220	Medium	0.4635	Higher	1.28%
Taizhou	0.3689	Lower	0.3665	Lower	0.4134	Medium	1.04%
Mean value	0.3704	Lower	0.4076	Medium	0.4645	Higher	2.08%

## Data Availability

The evaluation system’s data originate from the China City Statistical Yearbook, the City Statistical Yearbooks of 15 prefecture—level cities, the Water Resources Bulletin, the Environmental Status Bulletin, the EPS database (Global Statistical Data Analysis Plat-form), and the Baidu Index, with some missing data imputed.

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
