# Peer review of "Evaluation of the Governance Efficiency of Water Environmental Governance Efficiency in Yangtze River Delta from the Perspective of Multivariate Synergies"

_ijerph, 2022, doi:10.3390/ijerph19042347_

Round 1
Reviewer 1 Report
Dear Authors,
I found serious flows in the manuscript as:
- Subheadings are missing, you just left the original example there from the sample.
- Sometimes methodological parts are showing up in the chapters that I assume to be a "results" section.
- There is no result section.
- The discussion section is entirely missing (only appears in the heading).
Furthermore, there are numerous minor issues:
- abbreviations are just showing up in the text and not explained at first appearance, and some of them are in the upcoming tables but even these are not referred to,
- there are methods introduced in the sections where I believe that there are results and these methods are not discussed/introduced before at all,
- I have some problems with some phrases, too.
- It is not always clear how the calculations are made, as I read the text, I have a feeling that indexes should not show improvements but then, the overall index ratings are positive.
I think that English also needs to be improved.
I made numerous comments in the manuscript, in case the editorial office decides to reject and you decide to resubmit, maybe these comments can help.
Best regards, Reviewer X

Author Response
Modification Instructions of《Evaluation of the efficiency of water environmental governance in Yangtze River Delta from the perspective of multivariate synergies》
Dear reviewers and editor,
We earnestly appreciate it that your patient guidance is of great significance to our article. These suggestions and guidance are very important for the promotion of this manuscript. Although the current article still has some shortcomings, but under the guidance of reviewer and editor, I think we will be able to get the article promoted a lot. These time, after receiving your email, we made a modifying based on your suggestions. In addition, the modified version will be sent to you in the form of an attachment, and the particular modification instructions will be attached to the post.
Wish you have a really nice day!
Xiaoqiong Liu
2022-01-27
The detailed modification instructions were listed blow:
- Subheadings are missing, you just left the original example there from the sample.
Answer:We are very sorry that due to our oversight, some title changes were missed when we adjusted the reference template to the journal. In this revision, we double-checked and corrected these writing mistakes.
- Sometimes methodological parts are showing up in the chapters that I assume to be a "results" section.
Answer:Thanks to the reviewer for the reminder. This problem was caused by my personal habit of briefly describing the calculation process of the study before describing the results. To avoid this issue raised by the reviewer, we have revised several parts of the manuscript. For example, revised the first paragraph of 3.2 to “The water environmental governance efficiency index of cities in the Yangtze River Delta for each calendar year was calculated, and table 2 shows the representative indexes for the first and last years, as well as the midway year.”; revised the first paragraph of 4.1 to ”The results of the calculation of the efficiency of water environmental governance for different actors are shown in Figure 2.”;deleted the first paragraph of 4.2 “According to the formulas (8) and (9), the network matrix associated with the water environmental governance in the Yangtze River Delta is constructed, and Ucinet 6 is utilized to calculate various indexes to measure the overall structural characteristics and the individual centrality of the network.”;deleted the first paragraph 4.2.2 “The degree centrality, betweenness centrality, clossness centrality and system centrality of the relevant cities in the Yangtze River Delta were calculated using the SNA method, and the above four indexes were divided into different grades by the natural breakpoint method, and the actual situation in 2006, 2011 and 2017 was represented on the ArcGIS10.2 platform (Figure. 5)”.
- There is no result section.
Answer:In response to this challenge from the reviewer, and after discussion with other co-authors, I respond as follows. In most studies, authors prefer to organize the text in a paradigm of "Abstract, Keywords, Introduction, Methods and Materials, Results, Conclusions and Discussions". In my manuscript, considering the focus of the study and the balance of the length of each chapter, I divided the results section into two chapters, 3 and 4, which means that chapters 3 and 4 are actually my results, but they are not named as results and are not grouped into one chapter. Since I had seen this line in other journals, I tried to organize the content in a way that better suited the logic of my research. If the reviewer think that this style is not acceptable, please do not hesitate to let me know, and I will spare no effort to try to adjust and revise it if I have the opportunity to continue.
- The discussion section is entirely missing (only appears in the heading).
Answer:We apologize for the apparent weakness of our discussion section in the previous draft. Based on the reviewer's comment, we have divided the final chapter into two sections, Discussion and Conclusion, in this revision. The discussion section focuses on the significance of the study in this manuscript and the scientific and rational use of research methods in the manuscript. The specific modification are as follows:
5.1. Discussion
(1) This paper constructs a water environment governance efficiency evaluation system based on the DPSIR analysis model, which includes five dimensional layers of driving force, pressure, state, impact and response with a total of 20 indicators, and uses TOPSIS evaluation method and SNA method to conduct a comprehensive analysis of water environment governance efficiency in the Yangtze River Delta from 2006 to 2017. Theoretically, research on the evaluation of water environment governance at the scale of watersheds or economic zones can effectively enrich the scale of current water environment research. Meanwhile, research conducted from a multifaceted and collaborative perspective can help enrich the content of the traditional water environment governance research framework and help improve environmental governance-related theories. In practice, evaluating the efficiency of water environment governance in the Yangtze River Delta will be helpful for clarifying the roles and responsibilities of different actors and regions in the governance of the water environment, solving problems such as information asymmetry, promoting the coordinated governance of the water environment in the Yangtze River Delta and advancing the modernization of the national governance system and capacity.
(2) It is critical to build a scientific and adequate index system to track the efficiency of water governance. However, according to the current research literature, local and foreign researchers differ greatly in the selection of indexes, and no general consensus has been formed. The elements that influence water environment governance are complicated and diverse, such as the population, economy, society, and resources, all of which are interconnected. Based on the summary of relevant studies and the current situation in the Yangtze River Delta, this paper draws on the outstanding advantages of the DPSIR model in reflecting the interaction between socio-economic, environmental governance, and governance policies in water environment governance, and constructs an index system with 20 indexes at three levels, including the target level, dimension level, and index level, based on the principles of systematicity, objectivity, comparability, and accessibility.
(3) Water environment governance has the characteristics of externality, and the interests of government, enterprises, the public, and other subjects cross each other, water environment governance within urban clusters presents the characteristics of complexity, extensiveness and time lag, and has the characteristics of complex network structure of multiple subjects, multiple levels and multiple fields. SNA method is a quantitative analysis method that integrates the use of graph theory and mathematical models to study the association between social actors, and has been widely used in recent years to analyze the complex relationship structure between regions. The use of SNA method can be a good way to effectively portray the complex network structure of water environmental governance within urban clusters.
- abbreviations are just showing up in the text and not explained at first appearance, and some of them are in the upcoming tables but even these are not referred to.
Answer:We are sorry that due to our oversight, many of the abbreviations for research methods were not explained when they first appeared. In the revised draft, we have added explanations for each abbreviated term when it appears once to make up for the deficiencies in the previous draft.
- there are methods introduced in the sections where I believe that there are results and these methods are not discussed/introduced before at all
Answer:Because all three reviewers found some problems with the organization of the research methods in the manuscript, I have reorganized that section. In addition to the changes in comment 2 above, I also made some other modifications. First, the contents of the DPSIR analysis method section were trimmed so as to make the section more focused. Second, the content of the TOPSIS method section was improved, mainly by adding the introduction of the calculation process of EWM and MSD. In addition, after a reminder from another expert, I deleted the Thiel index in section 4.2. Finally, the calculation process of the social network analysis method has been elaborated. The specific modifications are shown in the red marked parts of the manuscript.
- I have some problems with some phrases, too.
Answer:From the reviewer's comments and notes, the main controversy is on the term "water environment treatment". According to the reviewer's comments, I reviewed a lot of literature on water environment and found that, as the reviewer said, the term "water environment treatment" is preferred when it comes to more specific problems such as water pollution, while when faced with more integrated water environment issues, the term "water environment governance" or "water environment management" is preferred. Our manuscript, then, undoubtedly lends itself to the term "water environment government". Once again, I would like to thank the reviewers for asking this question, which helped us to sort out the issues related to "water environment treatment" and "water environment governance" again. After sorting out this problem, we revised the manuscript for the words that needed to be replaced. The specific changes are shown in the revised version of the manuscript.
- It is not always clear how the calculations are made, as I read the text, I have a feeling that indexes should not show improvements but then, the overall index ratings are positive.
Answer:I apologize that the organization of the research methods section was somewhat problematic in the previous draft, and this may have created a barrier to understanding the content. In the revised manuscript, each research method was reorganized in the hope that it would help to understand the computational process. After seeing this comment from the reviewer, I reperformed one side of the calculation process for self-checking and found that my calculations were fine. As for the question raised by the reviewer that all indicators show a positive trend, I explain as follows: first, in the fourth column of Table 1, we indicate the question of whether each indicator is positive or negative. Second, at the very beginning of the data processing, we standardized all the indicators, this step is to eliminate the influence of the indicator scale on the calculation results, so the calculated results are positive in the end. But positive numbers do not mean that the result of water environmental governance is good, but the larger the value, the better the effect. Admittedly, the result show that the situation of water environment governance is indeed improving, which is closely related to the increasing awareness, technology, and policies of water environment governance in the study area.
- I think that English also needs to be improved.
Answer:I am very sorry that since we are neither native English speakers nor English majors, we do still have shortcomings in English writing. In the process of revision, we found that there were also tense mistakes, improper wording, and subject-predicate inconsistencies. In this round of revision, I invited undergraduate students majoring in English to help me check for grammatical mistakes and try to fix grammatical problems. For example, I revised "In terms of the spatial dimension, the water environment treatment efficiency of cities in the Yangtze River Delta is characterized by heterogeneous evolution. The Thiel index increased from 0.0023 in 2006, showing that the gap in treatment efficiency among cities further expanded" to "In terms of the spatial dimension, the water environmental governance efficiency of the cities in the Yangtze River Delta is characterized by heterogeneous evolution, and the gap in governance efficiency among cities has further expanded. " (in Line 324-326;). In Line 328, we revised "on the contrary" to "in contrast", etc. Since there are many corrections, I will not list them all here. The specific corrections can be found in the revised version of the manuscript. I hope it will meet the requirements of the reviewer. In case the reviewer thinks my language still has more places that need to be revised, I will invite a professional language touch-up agency to make touch-ups in order to meet the publication requirements of the reviewer and the journal.
In addition to the above comments, we also revised the comments in the marked-up version of the editor's feedback one by one, and the specific changes are shown in the marked-up version of the revised manuscript. (The content marked in red is revised)
Thanks again to the reviewers and editors for your profound and precise suggestions for this manuscript! If there is anything that does not meet your revision expectations, please do not hesitate to contact me and continue to give me your comments. I will be honored to cooperate with your revisions in order to fully revise the manuscript in the hope that it will meet the International Journal of Environmental Research and Public Health’s publication requirements.
Reviewer 2 Report
This is a very nice written paper about the methodology developed and applied to evaluate the water environment treatment status affected by many factors. Some minor changes could improve the quality of the paper.
The authors and affiliation are not provided under the title.
Avoid using the same terms many times in a sentence. For example "water environment treatment" has been repeated 3 times in a single sentence (lines 5-8)
In lines 66-67: “Existing relevant studies applying the DEA model tend to have too many single indexes, and not enough attention has been paid to the structural factors influencing water treatment efficiency”, please be more specific by giving examples of the “single indexes” and the “structural factors”
There are a lot of abbreviations (TOPSIS, DEA, DPSIR etc) which are not explained when they are first mentioned. Please provide the full words when you use abbreviations for the first time.
The abstract and introduction should report why the Yangtze River Delta was chosen for the case study. Section 2.1 gives this information but, in my opinion, this does not belong to this section. For example, the lines from 115 and below with the figure are the most relevant. Most of the content of 2.2.1 should be in the introduction or the discussion of the results. Section 2 should only contain what is relevant with the methods used, for example, the characteristics and details of the case area (but not why they were chosen), the indices used (and not why these indices were used – this should be explained in the discussion of the results). Please reorganize the text accordingly. Moreover, the titles of the subsections of 2.2 are weird and wrong. Please correct.
The legend in Figure 2 is not comprehensive. The sentence in lines 352-353 does not make sense. Moreover, “assess” may be a better verb than “measured”. The legend of the vertical axis in Figure 3 is too small to be read.
Most of the papers cited are in Chinese. These references cannot be comprehended by non-Chinese speaking people. Please use other sources written in English for reference.
Author Response
Modification Instructions of《Evaluation of the efficiency of water environment treatment in Yangtze River Delta from the perspective of multivariate synergies》
Dear reviewer and editor,
We earnestly appreciate it that your patient guidance is of great significance to our article. These suggestions and guidance are very important for the promotion of this article. Although the current article still has some shortcomings, but under the guidance of reviewer and editor, I think we will be able to get the article promoted a lot. These time, after receiving your email, we made a modifying based on your suggestions. In addition, the modified version will be sent to you in the form of an attachment, and the particular modification instructions will be attached to the post.
Wish you have a really nice day!
Xiaoqiong Liu
2022-01-27
The detailed modification instructions were listed blow:
- The authors and affiliation are not provided under the title.
Answer:Thanks for the reviewer’s reminder. In the first round of submissions, we submitted an anonymous manuscript and a manuscript with author information in the submission system, and it is possible that the editorial staff sent the anonymous manuscript to the reviewer. When submitting the revised manuscript, I will submit both types of manuscripts again to clarify the authors and our affiliation information.
- Avoid using the same terms many times in a sentence. For example, "water environment treatment" has been repeated 3 times in a single sentence (lines 5-8)
Answer:Thanks for the reviewer’s reminder. To avoid the same words appearing repeatedly in a sentence, we revised this sentence to " To clarify the roles and responsibilities of different regions and actors in water environment treatment, this study empirically analyzes the efficiency of water environment treatment in the Yangtze River Delta from 2006 to 2017 based on the DPSIR (Driving-Pressure-State-Impact-Response)-TOPSIS (Technique for Order Preference by Similarity to an Ideal Solution) method and the SNA (Social Network Analysis) method." in the new version of the modified manuscript.
- In lines 66-67: “Existing relevant studies applying the DEA model tend to have too many single indexes, and not enough attention has been paid to the structural factors influencing water treatment efficiency”, please be more specific by giving examples of the “single indexes” and the “structural factors”
Answer:This is a very interesting question. In the manuscript, we considered single engineering technical indicators as mainly those involving the water environment itself, such as nitrogen and phosphorus content in wastewater, while structural indicators were those involved in other subjects that would have an impact on the water environment (e.g., residents and enterprises), such as the water pollution Baidu index and industrial wastewater discharge. In the original draft, we did not elaborate the specifics of these two factors in more detail, and in this round of revision, we revised the sentence to "Existing relevant studies applying the DEA model tend to have too many single indexes, and the relevant indicator are mostly engineering technical indicators involving the water environment itself (e.g., nitrogen and phosphorus in wastewater), and not enough attention has been paid to the structural factors influencing water governance efficiency, lacking attention to residents, enterprises and other subjects. "
- There are a lot of abbreviations (TOPSIS, DEA, DPSIR etc) which are not explained when they are first mentioned. Please provide the full words when you use abbreviations for the first time.
Answer:We are sorry that due to our oversight, many of the abbreviations for research methods were not explained when they first appeared. In the revised draft, we have added explanations for each abbreviated term when it appears once to make up for the deficiencies in the previous draft.
- The abstract and introduction should report why the Yangtze River Delta was chosen for the case study. Section 2.1 gives this information but, in my opinion, this does not belong to this section. For example, the lines from 115 and below with the figure are the most relevant. Most of the content of 2.2.1 should be in the introduction or the discussion of the results. Section 2 should only contain what is relevant with the methods used, for example, the characteristics and details of the case area (but not why they were chosen), the indices used (and not why these indices were used – this should be explained in the discussion of the results). Please reorganize the text accordingly. Moreover, the titles of the subsections of 2.2 are weird and wrong. Please correct.
Answer:Thanks for the reviewer’s comment about research area and research method, I think these two parts do need to be revised, as the reviewer said. First, in the revised draft, we have adjusted the statement related to why the Yangtze River Delta was chosen as the study area to the introduction section, and the study area overview section only introduces the physical and socioeconomic characteristics of the study area. The research method section has also been adjusted visibly: the DPSIR analysis method section has deleted the reasons for the selection of indicators and only kept the content of what indicators were selected; the TOPSIS method section has been added, mainly to present more clearly how EWM and MSD were calculated; the SNA method section has also been improved. The specific changes are detailed in the annotated version of the manuscript.
- The legend in Figure 2 is not comprehensive. The sentence in lines 352-353 does not make sense. Moreover, “assess” may be a better verb than “measured”. The legend of the vertical axis in Figure 3 is too small to be read.
Answer:Thanks for the reviewer’s suggestions, I think it will help a lot to improve this manuscript. According to the reviewers' comments on the manuscript involving figures, we did realize that there were some problems such as incomplete legends and lack of clarity in the presentation of information. Therefore, in this round of revision, we reworked all the figures, and the specific changes are as follows. As for the sentence in lines 352-353, we changed it to “The results of the calculation of the efficiency of water environment treatment for different actors are shown in Figure 2.”
Figure 2. DPSIR subsystem evaluation indicators of Yangtze River Delta.
Figure 3. Synergistic actors water environmental treatment index of Yangtze River Delta.
Figure 4. Correlation index and network density of water environment treatment of Yangtze River Delta in 2006-2017.
Figure 5. Efficiency of the Water Environment Treatment Network of Yangtze River Delta in 2006-2017.
Figure 6. Individual centrality of the water environment treatment network of Yangtze River Delta in 2006-2017.
- Most of the papers cited are in Chinese. These references cannot be comprehended by non-Chinese speaking people. Please use other sources written in English for reference.
Answer:Thanks for the reviewer’s reminder of adding several English literatures. Since our research area is in China and more Chinese scholars have engaged in related research, it is difficult to avoid citing more Chinese literature in this manuscript. Of course, we acknowledge that the reviewer made a very pertinent suggestion that as a manuscript published in a foreign language journal, more English literature needs to be cited. Therefore, in the revision, we tried to refer to more international works, such as several English-language literatures listed below.
28 Svarstad, H., Petersen, L., Rothman, D., Siepel, H., Wätzold F. Discursive biases of the environmental research framework DPSIR. Land Use Policy. 2007, 25, 116-125.
29 Manjot, K., Kasun, H., Rehan, S. Investigating the impacts of urban densification on buried water infrastructure through DPSIR framework. Journal of Cleaner Production. 2020, 259:120897.
33 Hwang, C., Yoon, K., Multiple Attribute Decision Making. Lecture Notes in Economics & Mathematical Systems. 1981, 404: 287-288.
34 Chakraborty, S. TOPSIS and Modified TOPSIS: A comparative analysis. Decision Analytics Journal. 2022, 2: 100021.
42 Brandes, U., Hughes, H. Network Analysis: Methodological Foundations. Berlin, German: Springer-Verlage, 2005.
43 Pitts, F. A graph theoretic approach to historical geography. The Professional Geographer. 1965, 17: 15-20.
44 Pitts. F. The medieval river trade network of Russia revisited. Social Networks. 1978, 1: 285-292.
In addition to the above comments, we also revised the comments in the marked-up version of the editor's feedback one by one, and the specific changes are shown in the marked-up version of the revised manuscript. (The content marked in red is revised)
Thanks again to the reviewers and editors for your profound and precise suggestions for this manuscript! If there is anything that does not meet your revision expectations, please do not hesitate to contact me and continue to give me your comments. I will be honored to cooperate with your revisions in order to fully revise the manuscript in the hope that it will meet the International Journal of Environmental Research and Public Health’s publication requirements.
Round 2
Reviewer 1 Report
Dear Authors,
I see that you made many revisions in the article.
I am sorry to say but I (and I think all the journals do) need a Result section, so I still need to give a major!
My other concern is that there are still some shortcomings that make the manuscript difficult to judge.
I ask you to describe all indices, indexes, etc. in the Materials and Methods part of the manuscript.
E.g. you mention "degree centrality, betweenness centrality, closeness centrality, and system centrality" in lines 849-850 but I do not see these exact phrases elsewhere in the manuscript which is a serious problem.
Furthermore: Figure 2 is missing.
There are smaller problems, like you do not refer to Figure 6 that raises the question: Do you need it? Or did you describe it?
I still have problems with your "water environmental" related phrases, e.g. in lines 1188-1200 you have 3 distinct related phrases:
- water environmental governance
- water environmental capacity
- water environmental governance efficiency
- water environmental cooperative governance
I suggest having a separate Discussion chapter and do not forget to place your results in international and/or national context!
In the Conclusion part make sure that all conclusions are originating from your results and they are all conclusions and not results, nor discussions.
The reference list is having serious formatting issues (journal names need shortening, DOI is missing).
English still needs improvements, sometimes it blocks the understanding of the content.
The above mentioned comments also mean that there can be content-wise questions left in the manuscript but in its present form of the manuscript it is difficult to judge.
Best regards, Reviewer X

Author Response
Modification Instructions of《Evaluation of water environmental governance efficiency in Yangtze River Delta from the perspective of multivariate synergies》
Dear reviewer and editor,
We are much obliged to you for your constructive suggestions and patient guidance in the previous version, these suggestions and guidance are very important for the promotion of this article. Although the current article still has some shortcomings, but under the guidance of reviews and editors, I think we will be able to get the article promoted a lot. These time, after receiving your email, we made a second round of modifying based on your suggestions. In addition, the modified version will be sent to you in the form of an attachment, and the particular modification instructions will be attached to the post.
Wish you have a lovely day!
Best regards!
Xiaoqiong Liu
2022-02-07
The detailed modification instructions were listed blow:
- I ask you to describe all indices, indexes, etc. in the Materials and Methods part of the manuscript. E.g. you mention “degree centrality, betweenness centrality, closeness centrality, and system centrality” in lines 849-850, but I do not see these exact phrases elsewhere in the manuscript which is a serious problem.
Answer:I would like to thank the reviewer for your comment. According to the reviewer's request to describe all the indicators, I have added the details of the social network analysis method, including the introduction of eight additional indicators and the demonstration of their calculation process. The details are as follows, and I hope they will meet the requirements of the reviewers.
The SNA often exposes broad network properties like network density, network relatedness, network hierarchy, and network efficiency, as well as network characteristics like the degree centrality, betweenness centrality, closeness centrality, and system centrality for each node [37–44].
In equation (19), m denotes the number of actual relationships in the relationship matrix, n is the number of nodes, and n*(n–1) is the maximum number of relationships for any two nodes connected. The deeper the link between the nodes, the higher the value of this indicator, which shows the closeness of the network structure.
(19)
In equation (20), r is the number of unreachable pairs of points in the network, or the number of pairs of points that are not directly or indirectly associated between two nodes. This indicator represents the network structure’s robustness and vulnerability. The network structure has a high degree of association and is more robust when any two nodes are coupled with each other.
(20)
Inequation (21), s is the number of symmetrically reachable relationship pairs in the relationship matrix, while max(s) is the maximum possible symmetrically reachable relationship pairs in the network. This indicator measures the degree of asymmetric reachability among network nodes; a higher value implies that the hierarchical structure among nodes is more unequal, and a few nodes hold a dominant position in the network structure.
(21)
In equation (22), v is the number of redundant lines in the network and max(v) is the maximum possible number of redundant lines in the network. This indicator represents the number of redundant lines in the network. The lower the network efficiency, the more external connections the nodes have, the more overlaps there are, and the more stable the network is.
(22)
In equation (23), di is the number of direct relationships that exist between node i and other nodes, and n is the number of nodes. The bigger the value of this indicator, the more connections the node has with other nodes, and the larger the value of the indicator, the more central the node is in the network.
(23)
In equation (24), gjk is the number of relational paths that exist between nodes j and k, and gjk(i) is the number of relational paths between nodes j and k to pass through node i. The greater the value of the indicator, the more important node i‘s role in the network as an intermediary node, as it is at the node where many nodes are connected to each other.
(24)
In equation (25), dij represents the shortcut distance between nodes i and j. The value is taken as 1 if the two cities are directly connected and not indirectly connected through other cities. The higher the indicator’s value, the better the exchange path’s accessibility, suggesting the presence of direct connections between the node and many other nodes.
(25)
In equation (26), the centrality indicators are considered together using system centrality. (26)
- Figure 2 is missing.
Answer:Thanks to the reviewer for the reminder. We apologize for the absence of Figure 2 in the previous draft due to our oversight. In the new round of revision, we have added Figure 2 as follows.
Figure 2. DPSIR subsystem evaluation indicators of Yangtze River Delta.
- You do not refer to Figure 6 that raise the question: Do you need it? Or did you describe it?
Answer:We apologize that in the previous draft we did not mark in the manuscript what is relevant to Figure 6. Here we make some explanations and changes. Figure 6 depicts the four indicators of degree centrality, betweenness centrality, closeness centrality, and system centrality, which correspond to the second half of subsection 3.2.1. In the new revised version, we have labeled (Figure 6) after the corresponding content as a reminder.
- I still have problems with your “water environmental” related phrases, e.g. in lines 1188-1200 you have 3 distinct related phrases: water environmental governance, water environmental capacity, water environmental governance efficiency, and water environmental cooperative governance.
Answer:I'm sorry I didn't understand the question asked by the reviewer very well. Do you not understand the connotation of these words? Or is there confusion about the difference between the meanings of these words? Can you communicate further?
- I suggest having a separate Discussion chapter and do not forget to place your results in international and/or national context.
Answer:Based on the reviewer's comments, we split the original final section into two separate chapters, Conclusions and Discussions, and revised the contents of the Conclusions section to:
- Conclusions
A comprehensive analysis of the efficiency of water environmental governance provides some reference value for explaining the roles and duties of various actors and regions in water environmental governance, enhancing it in China, and perfecting the model. The following are the key conclusions of this study:
(1) The DPSIR analysis model is based on the integrated analysis of the dynamic mechanisms of water environmental governance, allowing for a more thorough assessment of the water environmental governance efficiency. The driving forces subsystem, as potential causes of risks and changes, often showed an increasing trend during the research period, and the driving effects on enhancing the Yangtze River Delta’s water environmental governance efficiency improved year by year. As the more direct pressure indicator, the pressure subsystem contributes the most to the water environmental governance efficiency. The state subsystem is a visual representation of water governance efficiency. Clearly, the index growth rate has a major impact on the efficiency of water governance. The impact subsystem index has been generally stable throughout time, indicating that the impact of water pollution on the ecological environment, economy, and society has remained relatively constant. The average annual growth rate of the response subsystem index is 4.06%, suggesting that the response measures from all sectors of society have played a substantial role in enhancing water governance efficiency. In general, the Yangtze River Delta’s water governance efficiency has increased steadily over time, and the governance efficiency of the water environment has risen from a low to a reasonable high level. It does, however, have a varied evolutionary spatial characteristic, and the gap in water environmental governance efficiency has grown even wider.
(2) The efficiency of different actors in terms of water environmental governance differs noticeably. The government’s water environmental governance efficiency was initially low in the early stages, but it has substantially improved since 2015. Enterprise governance efficiency has constantly improved, and it has increasingly assumed a leading role in the Yangtze River Delta’s water governance. The public’s role in the Yangtze River Delta's water environmental governance has been steadily increasing. However, sudden environmental occurrences with particular swings have an impact on their governance efficiency.
(3) At the regional level of coordinated governance, the Yangtze River Delta’s spatial association of water environmental governance is growing year by year, displaying the characteristics of a multi–city, multi–threaded, cross–regionally closely linked collaborative governance network, but with asymmetric structural characteristics. The degree of network incidence within each administrative region is high, especially in Zhejiang and Shanghai, which are relatively more central in the research area’s water environmental collaborative governance network, and water environmental governance coordination across provincial administrative boundaries still needs to be strengthened.
Revise the Discussions section to:
- Discussions
(1) To track the efficiency of water governance [24], it is important to develop a scientific and sufficient index system. However, according to the current research literature, local and international scholars choose indicators differently, and no general consensus has been reached. Water environmental governance is influenced by a complicated and diverse set of factors, including population, economics, society and resources, all of which are interconnected. On the basis of summarizing relevant research and combining the actual situation of the Yangtze River Delta, this paper draws on the outstanding advantages of the DPSIR model in reflecting the interaction among social economy, environmental governance and governance policies in water environmental governance. Based on the principles of comparability, comparability and accessibility, an evaluation index system with a total of 20 indicators is constructed at three levels, including the target layer, the dimension layer and the indicator layer. Compared with similar studies in China, we pay more attention to the structural factors of water environmental governance efficiency when selecting indicators, and comprehensively consider many indicators involved in contributors including the government, enterprises and the public (Table 1).
(2) Water environmental governance is characterized by externality, in which the interests of the government, enterprises, the public, and other stakeholders collide. Water environmental governance in urban clusters is characterized by complexity, extensibility, and time lag, as well as a complex network structure with various subjects, multiple levels, and multiple disciplines. The SNA method is a quantitative analysis method that combines graph theory and mathematical models to study the association between social actors, and it has been widely utilized in recent years to examine the complex relationship structure between regions. It is relatively rare in China to apply the SNA method to water environmental governance issues. From the research results, it can effectively depict the complicated network structure of water environmental governance inside urban clusters.
(3) This paper constructs a water environmental governance efficiency evaluation system based on the DPSIR analysis model, and uses the TOPSIS evaluation method and SNA method to conduct a comprehensive analysis of water environmental governance efficiency in the Yangtze River Delta from 2006 to 2017. Theoretically, studies on water environmental governance at the scale of watersheds or economic zones can effectively broaden the scope of current water environmental research. Meanwhile, multidimensional and collaborative perspective can serve to extend the content of the traditional water environmental governance research frameworks and help improve environmental governance–related theories. In practice, assessing water environmental governance efficiency in the Yangtze River Delta will be helpful for clarifying the roles and responsibilities of various actors and regions in water environmental governance, addressing issues such as information asymmetry, promoting coordinated governance in the Yangtze River Delta, and advancing the modernization of the national governance system and capacity.
(4) Based on the preceding conclusions and in conjunction with current conditions in the Yangtze River Delta, the following enlightenments are reached: ① It is important to keep the concept of systemic governance alive. The connection between socio–economic activity and the natural environment is reflected in the state of the water environment. The Yangtze River Delta’s water governance should consider regional economic development, population, natural resources, and environmental circumstances. ② To play a leadership and supervisory role in water protection, the government must further improve water governance efficiency. Meanwhile, the water environmental governance efficiency index should be fully utilized to ensure the right to know and supervise the actors involved, such as enterprises and the public, and to develop the water environmental governance system through multi–participation. ③At this stage, we need to pay more attention to the water environmental problems of small and medium–sized cities on the Yangtze River Delta’s outskirts, continue to improve the system and mechanisms of coordinated water environmental governance in the research area, and strive to solve the problem of insufficient water environmental cooperation across provincial administrative regions.
- In the Conclusion part make sure that all conclusions are originating from your results and they are conclusions and not results, nor discussions.
Answer:Thanks to the reviewer's reminder, in the new round of revisions, we removed the content that was not suitable to appear in the Conclusions section "Based on the preceding conclusions and in conjunction with current conditions in the Yangtze River Delta, the following enlightenments are reached: ① It is important to keep the concept of systemic governance alive. The connection between socio–economic activity and the natural environment is reflected in the state of the water environment. The Yangtze River Delta’s water governance should consider regional economic development, population, natural resources, and environmental circumstances. ② To play a leadership and supervisory role in water protection, the government must further improve water governance efficiency. Meanwhile, the water environmental governance efficiency index should be fully utilized to ensure the right to know and supervise the actors involved, such as enterprises and the public, and to develop the water environmental governance system through multi–participation. ③At this stage, we need to pay more attention to the water environmental problems of small and medium–sized cities on the Yangtze River Delta’s outskirts, continue to improve the system and mechanisms of coordinated water environmental governance in the research area, and strive to solve the problem of insufficient water environmental cooperation across provincial administrative regions.".
- The reference list is having serious formatting issues (journal names need shortening, DOI is missing).
Answer:Thanks for the reminder from the reviewer. According to the reminder from the reviewer, I mainly made the following changes to the format and content of the references: ① retrieved the abbreviations of the corresponding journal names of the cited literature, modified the names if there were abbreviations, and modified the format of the literature if there were no abbreviations, i.e., deleted the dot after the name of the literature; ② checked the DOI numbers of the cited literature and added them, there was also a situation that the DOI could not be retrieved, and we replaced the part of the (3) After the reviewer reminded us, we also checked the format of references again and found that the format of IJERPH required the initial letters of the titles of cited papers to be capitalized, and we also made corresponding changes.
- English still needs improvements, sometimes it blocks the understanding of the content.
Answer:I'm sorry that after the last round of revisions, there are still some problems with the English language. In this round of revisions, I invited a university English teacher who used to major in translation to guide me in the language revisions. This time, the English language revisions were mainly in the areas of correcting grammatical mistakes, improving the accuracy of wording, and enriching sentence patterns (see the revised version of the manuscript). However, in view of the fact that the time for revision was only one week and there were more contents to be revised, there might still be some deficiencies in the language. If the reviewers and editors are willing to give me another chance to improve this manuscript, I will invite a professional English language retouching agency to help revise the language. I hope the reviewers will give me a chance to revise.
In addition to the changes made to the comments in the peer review comments document above, I have also interpreted or revised the comments given in the annotated manuscript provided by the reviewers as follows:
- The problem of serial numbering of references in the main text.
Answer:In response to this question from the reviewer, I offer the following explanation: from the standard of IJERPH's reference number labeling, "," is used between two consecutive reference numbers, while three or more consecutive reference numbers use " —", and "," between two and more discontinuous reference numbers. I have followed this standard in my previous draft of the reference numbering, so I have not made any further adjustments in the new draft.
- Some issues where case is not uniform.
Answer:For example, the reviewer mentioned that only the initial letter of environmental in the chapter title "Water environmental Governance" was not capitalized, and in the new manuscript, this problem has been corrected. In the new manuscript, this problem has been corrected by changing the title of the chapter from "Water environmental Governance" to "Water Environmental Governance". In addition, after self-checking, I found that there was a problem with the inconsistency of the case of the title of the figure in the manuscript, and corrected it. In addition, in the format of IJERPH, the initial letters of Figure and Table are capitalized when figures or tables appear in the text.
- The problem of labeling serial numbers in chapter 2.2.2.
Answer:In my previous draft, the calculation of step (2) was divided into two parts, ① and ②. Maybe I was using ① and ② in the labeling way which led to some misunderstanding, so in the revised draft, I revised ① and ② to a and b, respectively, to avoid the barrier of understanding.
- Problems with understanding some phrases, such as "industrial undertaking areas".
Answer:For the sake of understanding, amend "industrial undertaking areas" to "areas where the secondary industry is concentrated".
- The reviewers also questioned some of the chapter titles.
Answer:The reviewer found the naming of the titles in Sections 4 and 4.2.1 problematic. After discussion with other co-authors, we also believe that the reviewer is justified in making this comment and that the current naming may indeed be misleading. Based on the reviewer's comments, I have revised the title of chapter 4.2 to " Analysis of Coordinated Water Environment Governance in Yangtze River Delta ".
- The naming of Figure 5 does not match the content of the figure.
Answer:Thank you for the reviewer's reminder. Based on the reviewer's comments, I have revised the title name of Figure 5 to "Correlation network efficiency of Yangtze River Delta’s water environmental governance in 2006—2017”
- The reviewers felt it was important to have a RESULTS section.
Answer:According to the request of the reviewer, I adjusted the organization of the chapters in the new round of revisions, and integrated chapters 3 and 4 into a single chapter. As a whole, the main part of the article is divided into chapters such as introduction, materials and methods, results, discussions and conclusions.
These are the revisions that I have made based on the comments made by the reviewer. If there is still anything that needs to be revised, please do not hesitate to contact me, and I will spare no effort to revise it carefully, hoping that this manuscript will meet the requirements of the reviewers and the IJERPH as soon as possible. Once again, I would like to thank the reviewers and editors for their willingness to give me the opportunity to revise, which is very significant for the quality of the manuscript!
